# Mechanical Properties of Polymer Coatings Applied to Fabric

**DOI:** 10.3390/polym12112684

**Published:** 2020-11-13

**Authors:** Serhiy Horiashchenko, Janusz Musiał, Kostyantin Horiashchenko, Robert Polasik, Tomasz Kałaczyński

**Affiliations:** 1Khmelnitskiy National University, st Institutskaya st. 11, 29016 Khmelnitskiy, Ukraine; tnt7@ukr.net (S.H.); gsl7@ukr.net (K.H.); 2Faculty of Mechanical Engineering, University of Science and Technology, Kaliskiego 7 Street, 85-796 Bydgoszcz, Poland; janusz.musial@utp.edu.pl (J.M.); tomasz.kalaczynski@utp.edu.pl (T.K.)

**Keywords:** polymer coating, fabric, polyethylene terephthalate (PET), tensile strength

## Abstract

The polymer film, formed on fabric, has a diverse resistance to impact (shear) forces during deformation. An original model of the capillary-porous structure of the fabric, partially filled with polymer, was presented and discussed in this paper. Polymer material fixing relations were developed, taking into account the fabric structure and changes of polymer temperature. Experimental studies were performed on three different materials: artificial leather SK-2, GOST 16119-70 (230 g/m^2^); genuine beef skin, GOST 939-75 (2.2 g/m^3^); and fabric denim, GOST 29298-2005 (225 g/m^2^). The value of mathematical model analysis deviation compared with the experimental value was approximately 12%. The obtained mathematical dependences were the basis for predicting the increase of the dimensional stability of garments by applying hot melt polymer to its surface. It is also possible to design new equipment for its implementation.

## 1. Introduction

In recent years, there has been a significant increase in the interest in fabrics coated with polymeric materials—this can be evidenced by a significant number of publications and reports from institutions dealing with the analysis of the coated fabrics market area. One such report [1] indicates expected development trends of the polyvinyl, polyurethane, polyethylene, and other (acrylic, nylon 6, nylon 6-6, PA, PC, PEEK, PBT, and polyethylene terephthalate (PET)) coated fabrics market. The authors of the study [1] anticipate an increase in the market of coated fabrics; from USD 16.3 billion in 2019 to USD 20.1 billion by 2024. Because of the favorable properties of coated fabrics, including flexibility, resistance to the effects of substances containing oils and fats, and increased abrasion resistance, they are becoming more and more often used in the light industry or for the production of everyday articles with increased functional properties (e.g., waterproof items).

Modern constructional polymers, especially composite materials, are often used in structural applications, where the ability to create properties such as stiffness, strength, or thermal conductivity makes them attractive compared with traditional engineering materials. Light and composite polymer materials were the subject of many research works and are used in various fields as support or load-carrying structures [2,3,4,5]. Materials engineering, also related to polymer composites formed on the basis of fabric and polymer coating, enables the production of materials with increased resistance to friction and wear [6,7]. Polymer coating deposition on a wide variety of engineering substrates has gained significant attention. Coatings are tailored to provide specific characteristics, such as wear, corrosion, chemical, and weathering resistance, to improve thermal conductivity; to provide electrical insulation; and, in the case of some polymers, to provide solid lubrication and low friction. Thermal spray and cold dynamic spray or simply cold spray are the main techniques employed to coat polymers onto substrates. The porous fabric absorbs part of the polymer coating, increases its durability, and changes its flexibility [8]. Polymer coated/modified structures’ and fabrics’ properties were the point of many studies and analyses [9,10,11,12].

The coatings with fluoropolymer resins rich in fluorinated ethylene propylene (FEP) and polytetrafluoroethylene (PTFE) are applied as anti-adherent coatings [2,13].

The process of applying a polymer coating has been studied and shown in many works, e.g., [3,5,7]. However, the polymer fixation process on fabric has not been studied in detail. This is because of the complexity of describing the process of polymer penetration into the tissue structure [12,14]. Typically, the polymer is applied to the fabric at an elevated temperature. After that, it cools and hardens.

Recent advances in polymer engineering, such as new emergency materials and new coating technologies, enable broader applications of polymeric coatings [7].

High bearing blended polymer coatings based on PEEK, PTFE, and ATSP materials show promising tribological behavior under aggressive contact conditions [15].

Among the main advantages of thermal spray are the following: the feasibility to coat large components of complex geometries, the use of powder as raw material (eliminating the use of solvents), and the possibility of the technique being applied in the field to coat and/or repair a fabric.

## 2. Phenomenon Principles and Theoretical Considerations

Nowadays, there is an increase in the availability of advanced techniques of applying polymers to the surface of fabrics. The most commonly used, spraying or padding, significantly change the structure and properties of the surface layer of fabrics, used for conventional and high-tech applications in life. In order to achieve high adhesion in the polymer composition deposited on the fabric surface, the gas-dynamic method is increasingly used [16,17,18]; this way provides, in increasing forms of stability, parts of garments made of fabric or leather by application of the direct method of stabilization of surfaces by polymer compositions, in particular polymer composition based on waste of polyethylene terephthalate (PET) [19]. Various ways are usually used to combine the gas and liquid phase transfer and functionality of equipment into coated textiles. In [19], the detailed process of applying a covering on a surface is presented, which uses waste of polyethylene terephthalate. Physico-mechanical properties are considered in [20,21,22]. The dependencies shown in [20] allow to determine the symptoms of the spherulitic structure destruction initiation. The method for determining the degree of natural polymer stretching can be used as a guideline in the design of equipment intended for the hardening of polymer sheets. The formation of a continuous liquid film that dampens the fabric was also not described by the wall interaction model. Liquid film properties have an influence on heat transfer and vaporization rates in high temperature applications. In the flooded case, heat transfer is modeled based on boundary layer correlations [20].

In the non-flooded regime, heat transfer can be modeled by considering correlations for individual drops impinging on a surface (Figure 1). The total heat transfer from the surface to a drop or film is shown in Figure 1 and was found from the following [23]:(1)$$Q=k_{t}h_{L}\left(T-T_{s}\right)\;t_{res}+Q_{g}$$
where

$h_{L}$—the liquid area in contact with the fabric;

$k_{t}=\mathrm{Nu}\cdot K$/d—the convective heat transfer coefficient, where Nu is the Nusselt number; d is equal to the drop diameter or, for flooded cases, to the film thickness; and K is the liquid thermal conductivity;

T—the liquid drop or film temperature;

$T_{s}$—the surface temperature on fabric;

$t_{res}$—the residence time for the liquid on the surface;

$Q_{g}$—the heat transfer from the gas (only relevant when a film is present).

**Figure 1 polymers-12-02684-f001:**
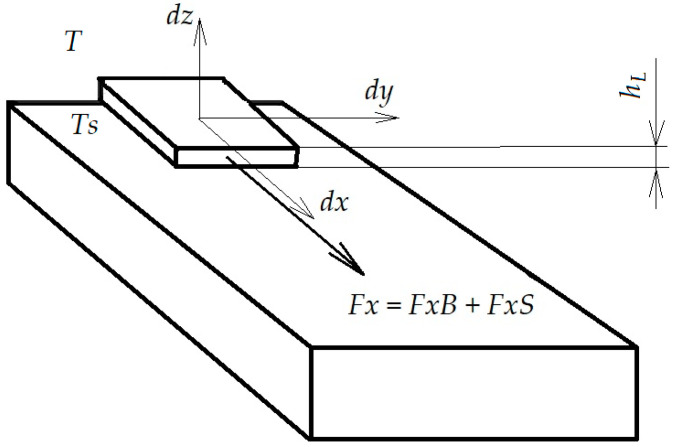
Model of applying polymer on the fabric surface.

Tasks were performed related to the formation in the liquid phase of additional internal elastic stresses in the formation of a coating of polymers at the stage of tensile formation of multimolecular parts. The polymer parts are reoriented according to the movement of the flow in intense motion and their hydrodynamic action leads to the formation of resistance to the general flow. Thermal spraying of polymer coatings can be used to protect the surface of parts from corrosion and mechanical impact [24,25].

The liquid coating process is a fluid in motion. The general macroscopic balance of forces is obtained by applying the principle of conservation of momentum to the volume in the liquid. The force acting on the volume was given by the rate of change of the momentum of the liquid surrounding it at any time, i.e., the flow of the momentum summed over the entire control surface, and the rate of change of the momentum in the volume (Figure 1).

The force Fx, acting in the x direction on a fluid element moving at the velocity of the polymer, is the sum of the force due to the weight of the volume element FxB and the force due to the stresses acting on it along the direction  x, FxS. So, Fx = FxB + FxS.

For an element of differential mass, ρ,  dx,  dy,  dz:(2)$$\rho dxdydz\left(\frac{du_{x}}{dt}\right)=\rho dxdydzgcos\beta +\left(\partial \tau_{xx}/\partial x+\partial \tau_{yx}/\partial y+\partial \tau_{zx}/\partial z\right)dxdydz$$
which, when permutated, is reduced to the following equation:(3)$$\rho du_{x}/dt=gcos\beta +\left(\partial \tau_{xx}/\partial x+\partial \tau_{yx}/\partial y+\partial \tau_{zx}/\partial z\right)$$
where

ux—the velocity of the liquid element in the direction x; 

*ρ*—the density of the polymer element; 

*g*—the acceleration of gravity; 

*β*—the angle that the polymer element makes with the axis x; 

t—time;

$\tau_{xx},\;\tau_{yx},\;\tau_{zx}$—stress components acting in directions *x*, *y*, and *z*.

The Navier–Stokes equation can be obtained by substituting the values of stress and velocity [26]. This is the equation of the element volume motion in the x direction and is used to analyze the hydrodynamics of the coating [10].
(4)$$\frac{u_{x}\partial u_{x}}{\partial x}+\frac{u_{y}\partial u_{x}}{\partial y}+\frac{u_{z}\partial u_{x}}{\partial z}+\frac{\partial u_{x}}{\partial t}=gcos\beta -\frac{\frac{1}{\rho \partial p}}{\partial x}+\eta /\rho (\frac{\partial 2u_{x}}{\partial x^{2}}+\;+\partial^{2}u_{x}/\partial y^{2}+\partial^{2}u_{x}/\partial z^{2})+\frac{1}{3}\frac{\eta }{\rho }\frac{\partial }{\partial x}\left(\frac{\partial u_{x}}{\partial x}+\frac{\partial u_{y}}{\partial y}+\frac{\partial u_{z}}{\partial z}\right)$$
where

$u_{x}\;,u_{y}\;,\;u_{z}$—the speed of the fluid in the directions x, y, and z; 

p—the pressure generated by the movement of the liquid element in the direction *x*;

*η*—the viscosity of the polymer.

With a simple analysis, the thickness of the film to be covered is a function of only five variables. These are the distance, speed, viscosity, density, and surface tension of the polymer to coat [11]. The movement of the polymer is considered stable and one-dimensional in nature, and the polymer is incompressible. For motion along the *x*-axis, where the *y*-axis is perpendicular to it, the equation decreases as follows:(5)$$\frac{\partial p}{\partial x}=\eta \frac{\partial^{2}u_{x}}{\partial y^{2}}+\rho g$$

The following boundary conditions are met: on the fabric, assume that the fluid is stationary, i.e., u=0, and the fluid flow velocity in the strip is the same as the velocity, $u=u_{0}$ for y=h.

The parallel flow model was used. Integrating the equation twice and applying the above boundary conditions for a Newtonian fluid, the velocity is obtained as follows:(6)$$u=u_{0}y/h+1/2\eta \left(dp/dx-\rho g\right)\left(y^{2}-h_{y}\right)$$

The total amount of fluid passing through the slit per unit length, per unit time, Q, is obtained by integrating the above speed between the limits $y=0,\;u=0\;i\;y=h,\;u=u_{0}$. The coating thickness, $h_{L}$, can be obtained by dividing Q by the velocity $u_{0}$.
(7)$$h_{L}=h/2-\left(1/12\eta u_{0}\right)\left(dp/dx-g\right)h_{3}$$

The pressure gradient with the surface tension of the coating fluid is obtained taking into account the balance of force:(8)$$dp/dx=-\sigma /2h_{2}$$
where

σ—the surface tension of the liquid.

Thus, the coating thickness obtained by taking into account the five above parameters is as follows:(9)$$h_{L}=h/2+\left(1/12\eta \right)\left(\sigma /2h_{2}+\rho g\right)h_{3}/u_{0}$$

The conclusion of the analysis is that, when applying the coating, the coating thickness consists of parts of the coating layer, some of which depend on the values directly related to the surface tension and gap, and others on the viscosity and velocity. However, this model does not take into account the parameters of the surface on which the polymer is applied. In our case, the fabric has a porous structure. Therefore, the task is to develop a model of polymer film formation and to study its fixation on the fabric surface.

## 3. Simulation of Polymer Layer Fixing on the Fabric

The scheme of displacement of the polymer layer fixed on the fabric is shown in Figure 2. The following assumptions were made: the polymer layer between the parts is an elastic body until the external forces exceed the critical value of molecular adhesion, and the fabric is absolutely solid and fixed. The action of the fixing force of the polymer layer $F_{f}$ should not exceed the value of the adhesive bond between the polymer and the fabric $F_{a}$ to maintain the adhesive fixation [10,11]. The critical value of the shear force is achieved at $F_{i}$ =$F_{f}$, at which the fixing forces are still active and the polymer remains an elastic body.

From [27], the value of the coefficient that takes into account the increase in the contact area between the polymer and the fabric can be obtained as follows:(10)$$k_{i}=1+\Pi_{i}\left(\frac{2h_{k}}{r_{k}}-1\right)$$
where

$\Pi_{i}$—the porosity of fabric.

The coefficient $k_{i}$ depends on the value of the porosity of the material, the average radius of the capillaries, and the depth of capillary penetration.

According to the microreological theory of adhesion, it takes time to fill the capillaries of the material of the parts with polymer. There is a dependence [28], which determines the effect of time on the depth of filling of the capillary of certain sizes, taking into account the viscosity of the liquid itself. However, this dependence does not have the form of a definite equation, it only notes that the value of the filling time is also affected by the amount of liquid itself. Using it, the coefficient that takes into account the time of filling the capillary with liquid can be determined according to the following formula:(11)$$K_{3}=\sqrt{\frac{Pt}{\eta }}\frac{2h_{k}\;}{r_{k}}$$
where

η—the viscosity of the polymer; 

t—the capillary filling time; 

$r_{k}$—the capillary radius; 

$h_{k}$—the depth of filling the capillary with polymer; 

P—external pressure.

This coefficient allows taking into account the delay of the formation of the contact plane in the formation of the adhesive interaction between the polymer and the fabric.

The adhesion force $F_{a}$ can relate to the work of adhesion. The specific work of adhesion can be calculated using the following equation [29]:(12)$$W_{a}=\sigma_{P}\left(1+\cos\Theta \right)$$
where

$\sigma_{P}$—reliable polymer tension; 

Θ—edge wetting angle of the solid surface.

The value of the adhesion energy can be determined by Formula (13), taking into account the plane of contact and the structure of the material:(13)$$E_{a}=W_{a}S_{P}k_{i}$$
where

$S_{P}$—the actual contact area.

Substituting the value (10) and (12) into (13), we achieve the following:(14)$$E_{a}=\sigma_{P}\left(1+\cos\Theta \right)S_{P}\left[1+\left(\frac{2h_{k}}{r_{k}}-1\right)\right]$$

Analysis (14) shows that the value of the adhesion energy depends on the physical properties of the polymer and the fabric, the depth of capillary penetration, and the contact area. The obtained equation allows quantifying the fixation energy.

According to the accepted model, the polymer layer deforms under the action of external forces and counteracts it with the force of adhesion. The nature of elastic forces is related to the forces of interatomic interaction in a fluid body. Adhesive forces cause the presence of intermolecular forces between the polymer and the fabric. In addition, the force of resistance to deformation affects the integrity of the coating. Therefore, in the future, a set of adhesion forces and deformation forces should be considered as a whole. Therefore, in the future, we consider the fixing force only as the sum of the forces of deformation of the fluid layer and the force of the adhesive interaction of the polymer and the fabric.
(15)$$F_{f}=F_{a}+F_{d}$$

The value of the fixation energy is characterized by the well-known Leonard–Jones equation, which is derived from the equation of work that must be applied to the elastic deformation of the polymer layer:(16)$$E_{d}=F_{f}\delta_{P}$$
where

$\delta_{P}$—the displacement of the polymer layer under the action of external forces.

As we consider the polymer to be an elastic body, the amount of energy of its fixation is determined from [16]. If the tangential strain stresses are characterized by the allowable value of adhesive strength under critical deformation conditions, we obtain the following:(17)$$E_{d}=\frac{\tau_{P}^{2}V_{P}}{2G_{P}}$$
where

$\tau_{P}^{}=\frac{F_{f}}{S_{P}}$—tangential stresses of deformation of the polymer layer; 

$V_{P}$—the volume of polymer; 

GP—the modulus of elasticity of the polymer.

The value of elastic deformation can be determined by the formulas of resistance of materials under external load [8]:(18)$$\delta_{P}=\frac{Fi_{2}h_{\Pi }{}^{3}}{3G_{P}J_{P}}$$
where

$h_{\Pi }$—the thickness of the polymer;

$J_{P}=\frac{\pi D_{P}^{4}}{32}$—the moment of inertia of the polymer

GP—the modulus of elasticity of the polymer.

According to [20,23], the value of the fixing energy is equal to the sum of the energies expended on the deformation of the polymer layer $E_{P}$ and destruction of the adhesive bond $E_{a}$, that is, $E_{f}=E_{P}+E_{a}$. Then, given that $F_{f}=F_{i}$ and substituting (16) в (17), the energy can be obtained from the following equation:(19)$$\sigma_{P}\left(1+\cos\Theta \right)\frac{\pi D_{P}^{4}}{32}K_{3}k_{i}+F_{f}\frac{32Fi_{2}h_{\Pi }{}^{3}}{3G_{P}\pi D_{P}^{4}}={\left(\frac{F_{f}}{S_{P}}\;\right)}^{2}\frac{V_{P}}{2G_{P}}$$

Using (13), (14), (16), and (18) in (19), the value of the fixing force can be determined:(20)$$F_{f}=\sqrt{\frac{\sigma_{P}\left(1+\cos\Theta \right)\frac{\pi^{3}G_{P}D_{P}^{6}}{32V_{P}}K_{3}{\left(1+\Pi_{i}\left(\frac{2h_{k}}{r_{k}}-1\right)\;\right)}^{}}{1-\frac{4\pi h_{\Pi }{}^{3}}{3V_{P}}}}$$

Analysis (20) shows that the fixing force depends on the properties of the material of the parts, while the polymer, the actual contact area, and the depth of capillary penetration depend on the application parameters.

The value of the expression $1-\frac{4\pi h_{\Pi }{}^{3}}{3V_{P}}$ approaches “1”, because the value of the second term of the expression is about 10^−10^. Thus, given that $S_{p}=\frac{\pi D_{P}^{}}{4}$, expression (20) can be written as follows:(21)$$F_{f}=\sqrt{\sigma_{P}\left(1+\cos\Theta \right)\frac{G_{P}S_{P}^{3}}{V_{P}}K_{3}\left(1+\Pi_{i}\left(\frac{2h_{k}}{r_{k}}-1\right)\;\right)}=\sqrt{K_{3}k_{i}\sigma_{P}\left(1+\cos\Theta \right)\frac{G_{P}S_{P}^{3}}{V_{P}}}$$

Using the values of parameters of drawing of the materials and volume of the polymer substituted in (20), the values of the fixing force can be defined precisely.

## 4. Experimental Studies

Surfaces of fabric: artificial leather SK-2, GOST 16119-70 (230 g/m^2^); genuine beef skin, GOST 939-75 (2.2 g/m^3^); and fabric denim, GOST 29298-2005 (225 g/m^2^) were chosen. The polymer composition based on waste of polyethylene terephthalate (PET) was used for experimental verification.

An experimental sample of equipment capable of spraying a certain amount of polymer is shown in Figure 3. The installation consists of the following: an electric motor (1), a frame (2), a crank mechanism (3), a tank for heating the polymer (4), a pump with a mixing chamber (5), and a spray (6) with a controlled nozzle (7). The spray breaks up and mostly vaporizes by the time it impinges at about 1 s after injection.

The tensile stiffness of the samples is determined in accordance with GOST 1497-84. The samples used in the tensile test are shown in Figure 4. The samples were tested on a computerized rupture complex, which could set different values of load, and measure the elongation of the sample (Figure 4). The average values of measurements from several samples of different thicknesses of the polymer film were obtained.

The values that allow determining the formula for the dependence of tensile strength on the thickness of the polymer coating z were calculated as follows:(22)$$4,8h_{p}-5L_{m}-0.085F_{r}+65.7=0$$
where

$h_{p}$—the experimental thickness of the polymer coating on fabric;

$L_{m}$—the value of the allowable elongation;

$F_{r}\;$—the allowable force.

The results of analytical calculations according to (21) are shown in Figure 5 for the volume of the polymer of 0.1$\;\times 10^{-4}$ m^3^, and the contact area was 0.04 × 0.04 m.

The obtained dependence allows predicting the amount of deformation during stretching of the material with a polymer coating depending on the thickness, which allows you to create new types of clothing based on the requirements of operation.

From Figure 6, it can be observed that the value of the fixing force of the textile increases with the increasing depth of capillary penetration, as well as with the decreasing film thickness. However, the reduction of the film to zero is impossible and, in reality, there is a required minimum thickness of the polymer layer, which is confirmed (see Figure 5). For $h_{k}=1.0\times 10^{-4}$ m, the film thickness $h_{\Pi }$ does not decrease and the fixing force $F_{f}$ does not change.

The temperature factor of the polymer also affects the value of the fixing force if the polymer is hot when it is applied. The value of the porosity of the material does not greatly affect the amount of fixation in this case, as it mainly deforms the upper layer of polymer. However, the higher initial temperature of the applied polymer causes the thinner coating layer. As the temperature increases, the polymer penetrates deeper into the porous structure of the fabric. The results of such exposure are shown in Figure 7; a camera with a special macro lens was used to qualitatively assess the adhesion when applying the hot melt adhesive to the fabric surface. Details with a polymer layer were photographed. The photos (Figure 7) show the most characteristic features of the applied polymers on the fabric surface.

As can be seen, the toast coating is formed at a polymer temperature of 80 °C (Figure 7a), and the coating is thinner at 90 °C (Figure 7b). The polymer has a partial penetration into the fabric at temperatures above 100 °C (Figure 7c). A good coating is formed at 120 °C (Figure 7d). A thin film of polymer with deep penetration into the porous structure is formed at temperatures above 150 °C (Figure 7e). However, the temperature of the polymer will not be allowed to exceed 210 °C, as the fabric will be destroyed.

The temperature of the polymer also affects the fixing force. The adhesive force value is small during the formation of the coating film, causing the polymer to penetrate into the porous structure of the material. After cooling, the force of fixation increases rapidly.

The values of the coating thickness are shown in Figure 8; for fabric denim, the penetration depth is almost linear (Figure 8, curve 1). The polymer layer depth is smaller owing to the more difficult penetration into the genuine beef skin (Figure 8, curve 2). This is because of the denser structure of the material. The values of fixing force for artificial leather are shown in Figure 8, curve 3. The average thickness of the coating (upper layer of coat) on the surface of the material is shown in Figure 8, line 4.

According to the obtained results, the amount of deformation can be determined analytically. The dependence of the allowable deformation of the polymer layer on the shear force is shown in Figure 9. For such small quantities, it can be linear.

The oscillogram for determining the limit value of the fixation force is shown in Figure 10. As can be seen, the polymer layer is destroyed after reaching the maximum value of the shear force.

Analysis of the dependence of the allowable deformation of the polymer layer on the shear force (Figure 10) shows that, with increasing shear forces, the deformation of the polymer layer increases linearly. However, deformation is possible only up to certain critical values. At $\delta_{P}>1.47\times 10^{-4\;}$ m, there is the destruction of a polymer layer.

## 5. Conclusions

The obtained results of applying the polymer to the surface of the material can be described by the theory of diffusion. This conclusion follows from the fact that the occurrence of mutual diffusion of macromolecules is observed in the boundary (upper) layer. As a result, an intermediate phase is formed. Diffusion theory involves the penetration of both liquid polymer molecules into the material and material molecules into the polymer as a result of their contact. These two processes lead to the disappearance of the boundary between the phases and the creation of a zone in which one polymer gradually passes into another. In this case, the adhesion is observed not as a superficial, but as a three-dimensional phenomenon. The strength of adhesion depends on the contact area, temperature, properties, and molecular weight of the polymers. The results of experimental studies show the possibility of increasing the fixation of the polymer, taking into account the porous structure of the material and application temperature.

The polymer layer on the fabric fixing model was developed and discussed. Theoretical analyses(modeling) were subjected to experimental verification for three materials: fabric denim, genuine beef skin, and artificial leather SK-2. The predicted polymer fixation process coincides with the experimental results; the comparison showed that the real (experimentally verified) bond strengths of the polymer with a given material were lower than the theoretical ones. This could result from the adopted modeling assumptions or the occurrence of distortions, e.g., material inhomogeneity or contamination and uncertainty of the polymer application rate, which was assumed to be constant during the simulation, while the polymer coating process of the material changed over time in the experiments. The obtained values showed that the instability of temperature fields is in the process of forming the coating on the fabric. Constant cooling of the polymer occurs during this process, which causes it to thicken quickly. Further, the loss of fluidity does not allow deep penetration into the material.

The process of filling capillaries for different materials was analyzed and calculated. The polymer is rigidly fixed in the fabric structure after cooling. This provides additional resistance to deformation forces of the polymer layer. The results showed a deviation of the mathematical model of 12% from the experimental one. The obtained mathematical dependences are the basis for predicting the increase in the shape stability of products by applying hot-melt polymers to their surface. It can also be used to give garments better wear resistance.

## Figures and Tables

**Figure 2 polymers-12-02684-f002:**
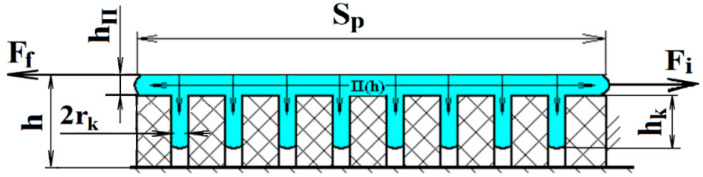
Model of fixing the polymer layer on the fabric.

**Figure 3 polymers-12-02684-f003:**
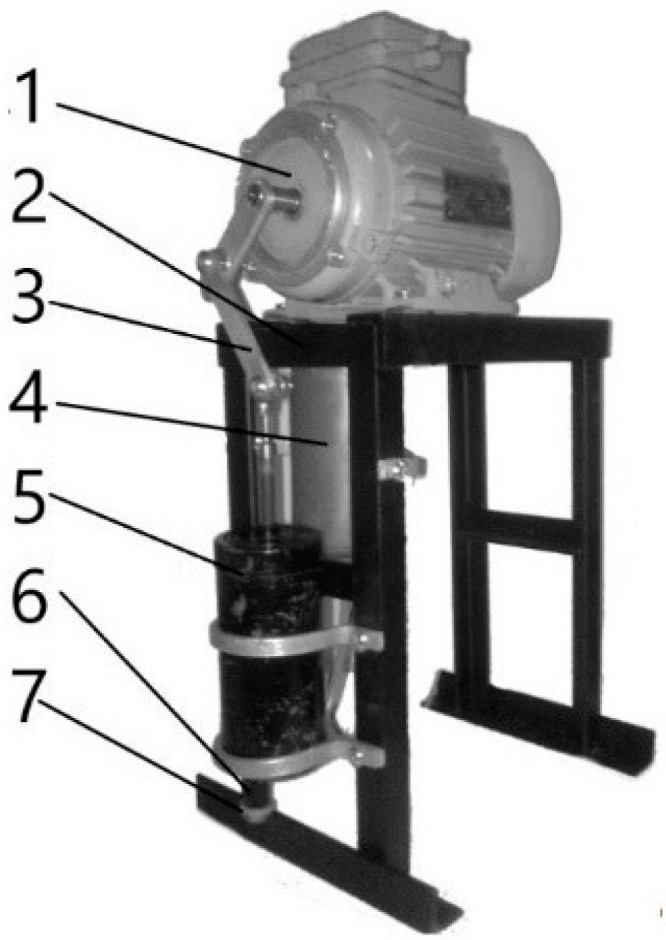
Experimental device for polymer cover application.

**Figure 4 polymers-12-02684-f004:**
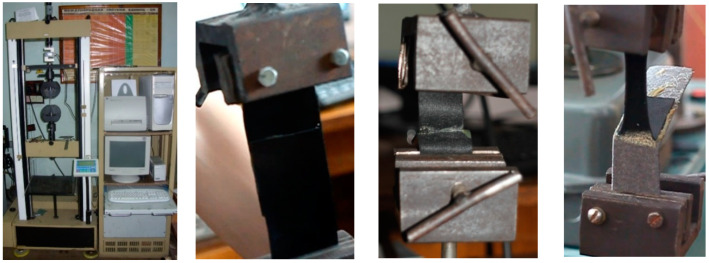
Polymer films on fabric tensile strength testing experimental set-up.

**Figure 5 polymers-12-02684-f005:**
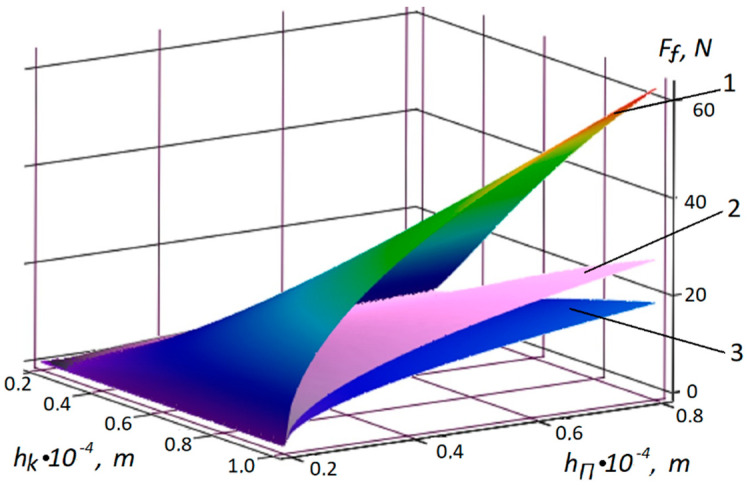
Dependences of the polymer coating thickness *h_k_* and of capillary penetration *h_π_* polymer on the tensile strength *F_t_* for (1) fabric denim, (2) genuine beef skin, and (3) artificial leather SK-2.

**Figure 6 polymers-12-02684-f006:**
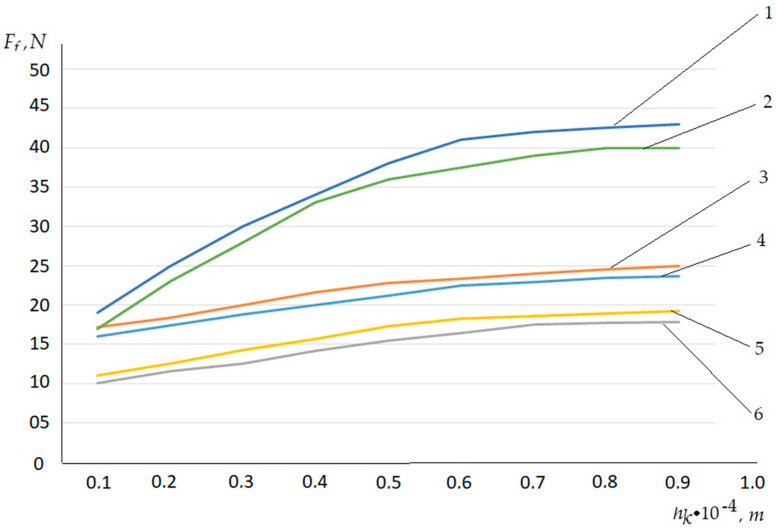
Dependences of the polymer film thickness film *h_k_* on fixing force *F_f_*. (1) Theoretical data for fabric denim, (2) experimental data for fabric denim, (3) theoretical data for genuine beef skin, (4) experimental data for genuine beef skin, (5) theoretical data for artificial leather SK-2, and (6) experimental data for artificial leather SK-2.

**Figure 7 polymers-12-02684-f007:**
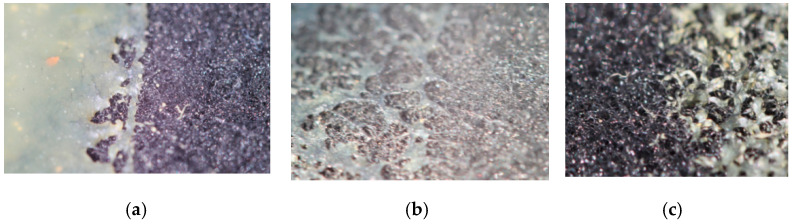
Experimental result of the influence of temperature of (**a**) 80, (**b**) 90, (**c**) 100, (**d**) 120 and (**e**) 150 °C on the formation of a polymer coating on the fabric surface.

**Figure 8 polymers-12-02684-f008:**
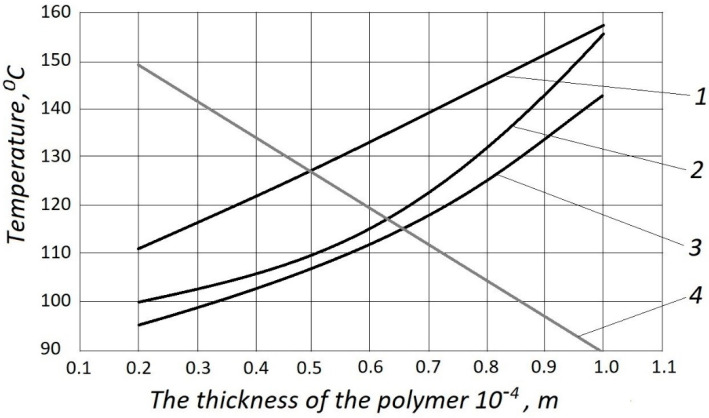
The relations between polymer coating thickness and temperature.

**Figure 9 polymers-12-02684-f009:**
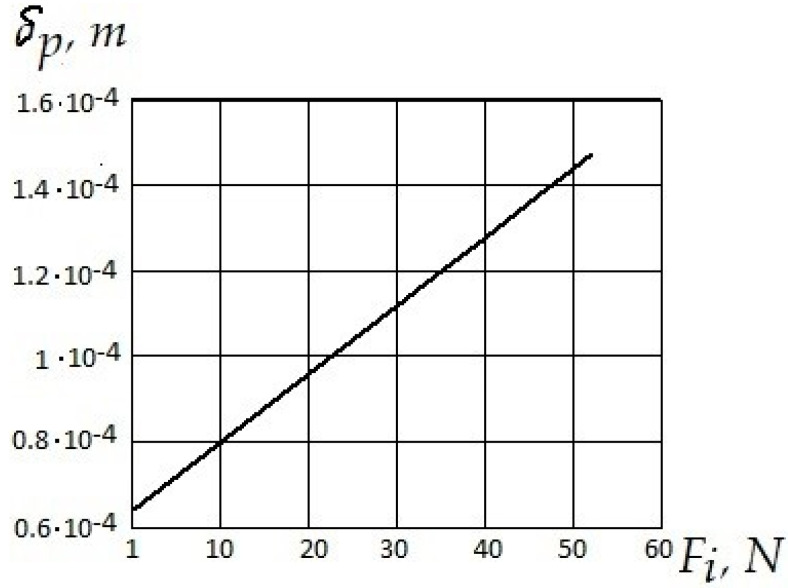
The dependence of the allowable deformation of the polymer layer on the shear force.

**Figure 10 polymers-12-02684-f010:**
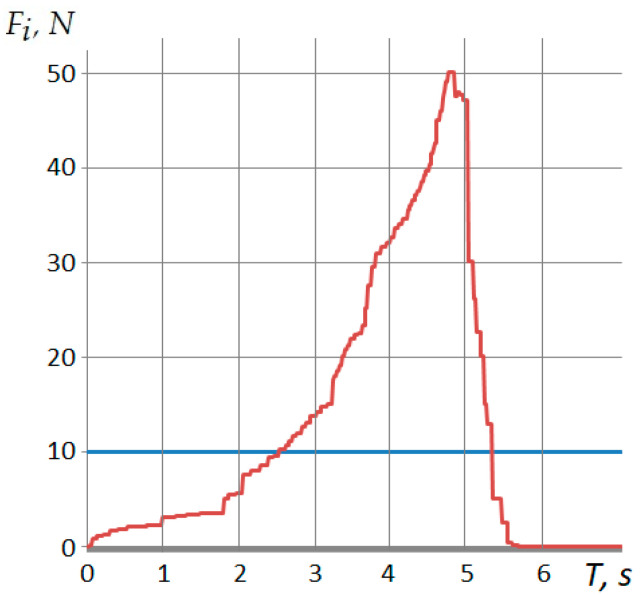
The oscillogram for determining the limit value of the fixation force.

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
