# Peer review of "Mechanical Properties of Polymer Coatings Applied to Fabric"

_polymers, 2020, doi:10.3390/polym12112684_

Round 1

Reviewer 1 Report

The manuscript addresses an interesting issue: the development of a theoretical model for the incorporation of polymers into the pores of textiles and the effect on the mechanical properties of the resulting composite. There is a lot of useful information. Regretfully grammatical and syntactic errors were not avoided and a careful recheck is necessary. It is very short and its structure needs to be thoroughly revised. The theoretical model of applying a polymer on the surface of a fabric should be moved from the introduction to a “results and discussion” section or a separate chapter. The Introduction section then should be extended in order to describe in more detail the uses of polymer-fabric composites and the previous research on the field (which polymers are used in combination with which textiles for what purpose). Critical information such as the name of the polymer and coating temperature is missing from the experimental section. In fact there is not a separate experimental section.

There are also many cases of carelessness and sloppiness:

1) The references are not quoted by numerical order.

2) References in Russian are not helpful in an English manuscript and should be withdrawn.  

3) Figure 5: The authors should incorporate the axis symbols (Ff, hk and hπ) to the legend.

4) In the legend of figure 6 the thickness of the polymer film is referred in contrast to the symbol hk corresponding to the depth of filling the capillary with polymer at the x axis legend. The scale of the x axis does not correspond neither to hk nor to hπ.

5) In the same legend the numerical order of the curves is wrong.

6) I m sorry but I do not understand the legend of Figure 8.

7) In the conclusion section the authors claim that “The predicted heat flux due combined gas and liquid phase transfer is in reasonable agreement with the experimental results”. I did not find them in the manuscript. A reference would be very helpful.

8) The references 16-18 are not marked in the manuscript.

Author Response

Comments and Suggestions for Authors

The manuscript addresses an interesting issue: the development of a theoretical model for the incorporation of polymers into the pores of textiles and the effect on the mechanical properties of the resulting composite. There is a lot of useful information. Regretfully grammatical and syntactic errors were not avoided and a careful recheck is necessary. It is very short and its structure needs to be thoroughly revised. The theoretical model of applying a polymer on the surface of a fabric should be moved from the introduction to a “results and discussion” section or a separate chapter. The Introduction section then should be extended in order to describe in more detail the uses of polymer-fabric composites and the previous research on the field (which polymers are used in combination with which textiles for what purpose). Critical information such as the name of the polymer and coating temperature is missing from the experimental section. In fact there is not a separate experimental section.

Changes in the layout of the article have been made, the "introduction" section have been enriched with new content (lines: from 25-39, 44-53, 56-57, 59-60, 63-64, 67-69, 74-85). Necessary information about samples and polymer, used in experimental verification have been added (lines: 266-268).

There are also many cases of carelessness and sloppiness:

  • The references are not quoted by numerical order.

Corrected.

  • References in Russian are not helpful in an English manuscript and should be withdrawn.  

Citation changes have been made.

  • Figure 5: The authors should incorporate the axis symbols (Ff, hk and hπ) to the legend.

Changes have been made.

  • In the legend of figure 6 the thickness of the polymer film is referred in contrast to the symbol hk corresponding to the depth of filling the capillary with polymer at the x axis legend. The scale of the x axis does not correspond neither to hk nor to hπ.

Changes have been made (figure 6).

  • In the same legend the numerical order of the curves is wrong.

Corrections have been made

  • I m sorry but I do not understand the legend of Figure 8.

New figures (7, 8 and 10) have been added.

  • In the conclusion section the authors claim that “The predicted heat flux due combined gas and liquid phase transfer is in reasonable agreement with the experimental results”. I did not find them in the manuscript. A reference would be very helpful.

New text fragments (lines from 313 to 341 and 349 to 350) have been added.

  • The references 16-18 are not marked in the manuscript.

Citation changes have been made.

Reviewer 2 Report

The current form of the manuscript indicates that the authors have no idea on how to construct a scientific journal paper in appropriate written English. A thorough revision is thus required for my reconsideration of the publication.

  1. Abstract

The abstract needs rewriting, since the current one is presented without coherence and in cumbersome English.

  1. Introduction

1) The references should be cited from 1, instead of 19. In addition, the authors should present the current references in Russian (e.g. Ref. #6) in English.

2) The introduction is not supposed to be presented in mathematical details. Instead, a coherent and concise literature review is required. It may help the authors by answering the following questions: Why are these works relevant? Which specific problems were addressed? How are the previous results related with the latest work? What are the outstanding, unresolved, research issues? Answering the questions leads to the novelty of the proposed work naturally.

3) Besides the literature review, some background information should also be presented. For example, the authors should introduce some examples of applications of constructional polymer, such as in porous materials [Fractals, 2020, 28 (2): 2050029] and petroleum engineering [SPE Production & Operations, 2018, 33(4): 770-783;].

  1. Simulation part

The theoretical part of the current introduction should be merged with the current simulation part, so that the new section can be titled as “Modeling of the problem”. In addition, cite a reference to an equation, unless it is proposed here originally.

  1. Experiment part

Although the results look “making sense”, the current form reads like a simple lab report at graduate level. The authors should dig deeper in the results by presenting some in-depth discussion.

  1. English writing

The current English writing is awfully cumbersome. The authors need to get a native English speaker for the thorough revision.

In a word, a revision according to my following comments is required before my reconsideration of publication.

Author Response

Comments and Suggestions for Authors

The current form of the manuscript indicates that the authors have no idea on how to construct a scientific journal paper in appropriate written English. A thorough revision is thus required for my reconsideration of the publication.

  1. Abstract

The abstract needs rewriting, since the current one is presented without coherence and in cumbersome English.

Changes and corrections have been made.

  1. Introduction

1) The references should be cited from 1, instead of 19. In addition, the authors should present the current references in Russian (e.g. Ref. #6) in English.

Citation changes have been made.

2) The introduction is not supposed to be presented in mathematical details. Instead, a coherent and concise literature review is required. It may help the authors by answering the following questions: Why are these works relevant? Which specific problems were addressed? How are the previous results related with the latest work? What are the outstanding, unresolved, research issues? Answering the questions leads to the novelty of the proposed work naturally.

Changes in the layout of the article have been made, the "introduction" section have been enriched with new content (lines: from 25-39, 44-53, 56-57, 59-60, 63-64, 67-69, 74-85).

3) Besides the literature review, some background information should also be presented. For example, the authors should introduce some examples of applications of constructional polymer, such as in porous materials [Fractals, 2020, 28 (2): 2050029] and petroleum engineering [SPE Production & Operations, 2018, 33(4): 770-783;].

Thank you for your valuable attention; it has been included in the "introduction" chapter.

  1. Simulation part

The theoretical part of the current introduction should be merged with the current simulation part, so that the new section can be titled as “Modeling of the problem”. In addition, cite a reference to an equation, unless it is proposed here originally.

New sections, titled “Phenomenon principles and theoretical considerations” and “Simulation of polymer layer fixing on the fabric “ have been added.

  1. Experiment part

Although the results look “making sense”, the current form reads like a simple lab report at graduate level. The authors should dig deeper in the results by presenting some in-depth discussion.

New text fragments (lines from 313 to 341 and 349 to 350) and figures (7, 8, 10) have been added. “Conclusions” section have been enriched with new content.

  1. English writing

The current English writing is awfully cumbersome. The authors need to get a native English speaker for the thorough revision.

 Several language changes have been made. Authors will ask editor for language correction.

In a word, a revision according to my following comments is required before my reconsideration of publication.

Round 2

Reviewer 1 Report

Plagiarism detected.

Author Response

The team of authors apologizes for this situation - see explanation below.

We (authors) found that a draft has been uploaded (instead of final version), in which we have placed the content of the report (Grand View Research. Coated Fabrics Market Analysis By Product (Polymer, Rubber, Fabric Backed Wall Coverings) By Application (Transportation, Protective Clothing, Industrial, Furniture) And Segment Forecasts To 2020. San Francisco: Grand View Research, Inc, 2014. ISBN 9781680381405) as a hint for us to write an introductory sentence, as suggested by the reviewer. This fragment should not to be included in the content - by mistake (the team works in different countries, the current situation and time pressure caused this omission) - it was not removed from the uploaded version. Another sentence was also not removed (from the article https://doi.org/10.1016/j.aiepr.2018.05.001) and it was not included in the citations.

Necessary corrections have been made in the attached file in the track changes mode.

Citations have been complemented.

Reviewer 2 Report

Basically, my comments have been addressed. In ref. 4, the page is lost, which is 2050029.

Author Response

The team of authors would like to thank you for your favorable review.

It was decided to supplement the article with new content, which resulted in a change of citations. One scientific achievement that is not available in the English language was abandoned.

We (authors) found that a draft has been uploaded (instead of final version), in which we have placed the content of the report (Grand View Research. Coated Fabrics Market Analysis By Product (Polymer, Rubber, Fabric Backed Wall Coverings) By Application (Transportation, Protective Clothing, Industrial, Furniture) And Segment Forecasts To 2020. San Francisco: Grand View Research, Inc, 2014. ISBN 9781680381405) as a hint for us to write an introductory sentence, as suggested by the reviewer. This fragment should not to be included in the content - by mistake (the team works in different countries, the current situation and time pressure caused this omission) - it was not removed from the uploaded version. Another sentence was also not removed (from the article https://doi.org/10.1016/j.aiepr.2018.05.001) and it was not included in the citations.

Necessary corrections have been made in the attached file in the track changes mode.

Round 3

Reviewer 1 Report

OK